# Reduced Titers of Circulating Anti-SARS-CoV-2 Antibodies and Risk of COVID-19 Infection in Healthcare Workers during the Nine Months after Immunization with the BNT162b2 mRNA Vaccine

**DOI:** 10.3390/vaccines10020141

**Published:** 2022-01-18

**Authors:** Luca Coppeta, Cristiana Ferrari, Giuseppina Somma, Andrea Mazza, Umberto D’Ancona, Fabbio Marcuccilli, Sandro Grelli, Marco Trabucco Aurilio, Antonio Pietroiusti, Andrea Magrini, Stefano Rizza

**Affiliations:** 1Department of Occupational Medicine, University of Rome Tor Vergata, 00133 Rome, Italy; luca.coppeta@ptvonline.it (L.C.); cristiana.ferrari@ptvonline.it (C.F.); giuseppina.somma@ptvonline.it (G.S.); andrea.mazza@ptvonline.it (A.M.); umberto.dancona@ptvonline.it (U.D.); andrea.magrini@uniroma2.it (A.M.); 2Department of Experimental Medicine, University of Rome Tor Vergata, 00133 Rome, Italy; fabbio.marcuccilli@ptvonline.it (F.M.); sandro.grelli@ptvonline.it (S.G.); 3Department of Medicine and Health Sciences “V. Tiberio”, University of Molise, 86100 Campobasso, Italy; marco.trabuccoaurilio@unimol.it; 4Departmental Faculty of Medicine, University Unicamillus, 00131 Rome, Italy; antonio.pietroiusti@unicamillus.org; 5Department of System Medicine, University of Rome Tor Vergata, Viale Oxford 81, 00133 Roma, Italy

**Keywords:** COVID-19, SARS-CoV-2, healthcare workers, vaccine, BNT162b2 mRNA vaccine, anti-S-RBD antibodies

## Abstract

The severe acute respiratory syndrome coronavirus 2 (SARS-CoV-2) pandemic has had a tremendous impact on health services; hundreds of thousands of healthcare workers (HCWs) have died from coronavirus disease 2019 (COVID-19). The introduction of the BNT162b2 mRNA vaccine in Italy provided recipients with significant protection against COVID-19 within one to two weeks after the administration of the second of the two recommended doses. While the vaccine induces a robust T cell response, the protective role of factors and pathways other than those related to memory B cell responses to specific SARS-CoV-2 antigens remains unclear. This retrospective study aimed to evaluate the determinants of serological protection in a group of vaccinated HCWs (*n* = 793) by evaluating circulating levels of antiviral spike receptor-binding domain (S-RBD) antibodies during the nine-month period following vaccination. We found that 99.5% of the HCWs who received the two doses of the BNT162b2 vaccine developed protective antibodies that were maintained at detectable levels for as long as 250 days after the second dose of the vaccine. Multivariate analysis was performed on anti-S-RBD titers in a subgroup of participants (*n* = 173) that were evaluated twice during this period. The results of this analysis reveal that the antibody titer observed at the second time point was significantly related to the magnitude of the primary response, the time that had elapsed between the first and the second evaluation, and a previous history of SARS-CoV-2 infection. Of importance is the finding that despite waning antibody titers following vaccination, none of the study participants contracted severe COVID-19 during the observational period.

## 1. Introduction

Health services are recognized as critical components in the overall response to the severe acute respiratory syndrome coronavirus 2 (SARS-CoV-2) pandemic. According to the World Health Organization (WHO), between 80,000 and 180,000 healthcare workers (HCWs) died after developing coronavirus disease 2019 (COVID-19) between January 2020 and May 2021 [1]. In many countries, the availability of qualified HCWs has been extremely curtailed; this has resulted in major disruptions to essential health services. Thus, vaccination of HCWs to prevent occupational infection and maintain appropriate healthcare services is a crucial global priority [2].

On 21 December 2020, four COVID-19 vaccines were authorized for use in Italy by the Italian Medicines Agency (AIFA). Among the first of this group was the BNT162b2 mRNA vaccine developed by BioNTech SE and Pfizer, Inc., New York & Mainz, Germany (tradename Comirnaty^®^) [3]. The BNT162b2 vaccine contains a nucleoside-modified RNA that encodes the spike (S) protein of SARS-CoV-2 [4]. The receptor-binding domain (RBD) of the viral S protein is the main target of antibodies that are capable of coronavirus neutralization. The viral S protein is a large transmembrane protein that plays a crucial role in promoting virus-mediated infection of target cells. Antibodies produced by subjects immunized with the BNT162b2 vaccine are directed against the RBD contained within the S1 domain, thereby hindering virus entry into cells [5]. Several studies have documented that anti-spike and anti-RBD polyclonal IgGs provide proportional neutralization of SARS-CoV-2 [6]. Other antibodies that develop in response to natural SARS-CoV-2 infection (i.e., anti-N nucleoprotein IgG) have been detected in convalescent plasma but not in vaccinated subjects; these antibodies are largely non-neutralizing [7,8].

Full protection against SARS-CoV-2 infection can be achieved within one to two weeks after the administration of the second of the two 30 μg doses of BNT162b2. Results from several reports revealed that a detectable level of protection might also develop in subjects who received a single dose of the vaccine [9,10,11]. The duration of protection against SARS-CoV-2 infection in both convalescent and vaccinated populations remains a subject of significant research interest, given that this response can be attributed, at least in part, to humoral immunity [12].

While both convalescent and vaccinated subjects show evidence of immunological memory in the form of specific B and T cell responses, clear laboratory correlates for protective immunity have not yet been defined [13]. To date, levels of circulating anti-SARS-CoV-2 neutralizing antibodies correlate most strongly with protection against both repeat infections and breakthrough infections in vaccinated subjects [14]. Some reports have indicated that vaccination results in higher anti-S IgG titers and stronger serum neutralization activity than is typically detected in response to natural infection [15]. Although the vaccine formulations do induce robust T cell responses, it is not yet clear whether immunological protection results from mechanisms that are independent of the actions of memory B cells [3,6,8,9].

The strong immunological response initially elicited by vaccination appears to wane over time in association with diminished circulating antibody titers and reductions in the number of memory T and B cells. However, it is not yet clear whether the vaccine-mediated protection against infection, severe disease, and death also wanes over time in association with diminished titers of anti-S IgG and reductions in serum neutralizing capacity. Results from a recent survey revealed that the decrease in vaccine effectiveness against SARS-CoV-2 infections over time could be attributed to waning immunity rather than the development of a viral escape mutant (i.e., the Delta variant) [16].

At this time, we have only a limited understanding of the duration of vaccine-induced protection against COVID-19. These findings may be crucial for the ongoing assessment of the risk of breakthrough infection among vaccinated HCWs who are routinely exposed to SARS-CoV-2 and for evaluation of the need for vaccine booster doses. Thus, this study aimed to evaluate the determinants of serological protection in a group of immunized HCWs via an exploration of the levels of circulating anti-S-RBD antibodies during a nine-month period following vaccination.

## 2. Materials and Methods

We performed a retrospective evaluation of data that are collected routinely at our hospital facility. The study was approved by the Ethics Committee of Polyclinic Rome “Tor Vergata” (184/2021). Our study enrolled 793 HCWs who were actively employed at the Polyclinic “Tor Vergata” and who received the complete vaccination series, including two doses of 0.3 mL of the BNT162b2 mRNA vaccine by 15 March 2021, after providing informed consent.

Demographic data and confirmation of SARS-CoV-2 infection (either before or after vaccination) were collected from the Occupational Medicine Service of the University of Tor Vergata. The following data were collected for each study participant: age, sex, job task, night shift status, date of the second vaccine shot, serum level of anti-S-RBD antibodies, date of serological evaluations, and the number of serological evaluations. Participants who worked a 12 h shift on four to seven nights per month, followed by two days off (*n* = 435), were identified as night shift workers (NSWs). Participants who did not work any night shifts (*n* = 358) were identified as daytime workers (DWs). The number of days that elapsed between completed vaccination and serological evaluation was calculated for each participant.

During the study period, each participant underwent one or more venipunctures to collect 10 mL blood samples that were used to determine the serum levels of anti-S-RBD antibodies. The Roche “Elecsys^®^” kit, which is designed for in vitro quantitative determination of antibody levels, was used to detect and determine the levels of high-affinity anti-S-RBD antibodies in each blood sample; positive results were identified as levels ≥0.8 U/mL, as per the manufacturer’s instructions. Each subject enrolled in our study underwent at least one evaluation.

A subset of participants (*n* = 173) underwent two serological evaluations during the study period. We recorded the date of the first and second evaluations and calculated the changes in anti-S-RBD antibody titers detected over the intervening time period.

### Statistical Analysis

Quantitative data are presented as means ± standard deviations (SDs). Categorical variables are presented as the number (percentage) of study participants. We calculated the mean antibody titer against viral S-RBD in the group of HCWs who completed the full two-dose vaccination cycle. We also calculated the changes in anti-S-RBD titers exhibited by participants who underwent two serological evaluations during the study period. Linear regression analysis was performed to identify any associations linking the observed decrease in antibody titers in this latter group with night shift status, previous natural viral infection, first antibody titer, age, and gender. However, because age and male gender did not contribute to the risk of the primary outcome, our multivariable models were limited to the other aforementioned variables. All analyses were performed using IBM^®^ SPSS^®^ Statistics (version 26), with a level of significance set at *p* < 0.05.

## 3. Results

We enrolled 793 HCWs (mean age 43.9 ± 11.3 years) who worked in various areas of the hospital. Of this cohort, 262 of the participants were physicians (33.0%), 269 were nurses (33.9%), and 262 (33%) were identified as other healthcare professionals. Table 1 includes descriptive characteristics of the study participants and includes both vaccination and serological information. Figure 1 summarizes the main clinical characteristics of the study population.

All participants received two doses of the BNT162b2 mRNA vaccine; a total of 25 (3.5%) had been previously diagnosed with COVID-19. Of them, 22 had COVID-19 with mild to moderate symptoms, while 3 had an asymptomatic disease.

Circulating anti-spike S-RBD antibodies were detected in 99.75% of the study population (i.e., in 791 of the 793 participants). Six hundred and twenty of the participants underwent a single serological evaluation during the study period, while one hundred and seventy-three participants were evaluated twice.

The interval between the final vaccination and the serological evaluation ranged from 13 and 253 days (mean 122.9 days). The serological evaluation was performed at >90 days and >150 days after the second dose of the vaccine in 403 (50.8%) and 164 (20.7%) of the participants, respectively. Twenty-five of the study participants had been diagnosed with COVID-19 before vaccination, while thirteen were diagnosed with COVID-19 after completion of the two recommended doses. Interestingly, none of the study participants were diagnosed with COVID-19 during the interval between the two doses.

Our findings document an overall reduction in the mean antibody titer over time (r = −0.434, *p* < 0.001, Figure 2). Specifically, our findings reveal a mean antibody titer of 1602.3 U/mL in participants who were evaluated within the first 90 days, and 1035.9 U/mL among those evaluated between 91 and 150 days after completion of the two-dose vaccination. As anticipated, we detected even lower mean antibody titers (724.4 U/mL) among those who were evaluated >150 days after vaccination (*p* < 0.01). Overall, female participants exhibited higher antibody titers than their male counterparts (1602.3 versus 1305.9 U/mL, respectively; *p* < 0.01).

In the 173 participants who underwent two serological evaluations, the sampling time ranged from 11 to 183 days (mean 92 ± 42). Figure 3 displays the dynamic changes in antibody titers from the first to second serological evaluations. Among them, the observed decreases in antibody titers were significantly related to previous SARS-CoV-2 infection (*p* < 0.05), the number of days between the two serology assessments (*p* < 0.01), night shift status (*p* < 0.05), and the magnitude of the first set of serological results (*p* < 0.01). Interestingly, these decreases did not correlate with age or sex. Of note, we excluded the four subjects who contracted COVID-19 between the two serological evaluations because the acute infection may have had a natural booster effect.

To avoid the possibility of confounding interactions between the study variables, we performed a multiple regression analysis. Our model included the factors identified in the univariate analysis that were significantly associated with diminished antibody titers. The results of the multiple regression analysis reveal that diminished antibody titers observed in the second serological evaluation were significantly associated with the magnitude of the results of the first evaluation, the time that elapsed between the first and the second evaluation, and a previous diagnosis of COVID-19. By contrast, diminished antibody titers were not associated with night shift status (Table 2).

## 4. Discussion

Although they were first introduced twelve months ago, the duration of protection afforded by COVID-19 vaccines as well as the need for and timing of booster doses remains unclear. These concerns continue to be major public health issues largely because a feasible, working correlate of protection against COVID-19 remains to be established. While the BNT162b2 mRNA vaccine clearly elicits strong memory and effector T cell responses [17], the roles of these responses in providing vaccine-mediated protection independent of memory B cells have not been fully addressed. Thus, the extent of vaccine-mediated immunity is largely based on its effectiveness in real-world studies combined with the detection of neutralizing antibodies (i.e., those that target the RBD region of the SARS-CoV-2 S protein) [17,18]. Previous studies have documented a strong correlation between serum levels of neutralizing antibodies and vaccine efficacy at preventing symptomatic and severe disease [8,10]; these results also suggest that the threshold of protection against severe disease could be lower than that needed for protection against symptomatic infection. While serum samples from vaccinated individuals are more effective than convalescent plasma at neutralizing SARS-CoV-2 viruses in vitro [7], the kinetics of the relevant antibody response following vaccination have not been fully assessed. To date, detectable levels of antiviral antibodies have persisted for up to 8 months after acute SARS-CoV-2 infection and up to 6 months after completion of the two-dose mRNA vaccine regimen [16]. Previous studies focused on patients who recovered from SARS-CoV-2 infection revealed that ~22% exhibited a rapidly waning antibody response and 12% had no detectable response whatsoever [18]. By contrast, vaccination of immunocompetent HCWs in our study appeared to elicit stronger and more persistent serological responses compared to findings reported in the aforementioned published studies.

Persistence of protection after vaccination is currently a critical healthcare issue, as is the need to evaluate the efficacy of a booster dose for subjects at high risk of COVID-19 complications and those at high risk of exposure, including HCWs. The results of our study clearly demonstrate detectable levels of anti-S-RBD in nearly all participants; these levels remain in the detectable range for up to nine months after vaccination. We also found that circulating antibody titers wane at rates that were directly related to the magnitude of the first serological response evaluated as well as the time elapsed between repeated serological evaluations. This latter result indicates that the humoral response to SARS-CoV-2 tended to wane over time. Several studies focused on long-term immune responses demonstrated that, after an initial exponential decay, antibody half-lives typically stabilize to ≥10 years. Depending on when this transition occurs, vaccinated subjects with waning neutralizing antibody titers may become susceptible to mild infection while remaining protected against severe disease [19]. In our study cohort, we identified 13 participants who became infected with SARS-CoV-2 after the second of the two vaccine doses. Of note, four of these individuals contracted SARS-CoV-2 infection during the time elapsed between the two sequential serological evaluations. These data confirm findings documenting the real-world effectiveness of COVID-19 vaccination and the importance of achieving a high rate of vaccination coverage among HCWs. This point remains critical, as hospitals and clinical care centers struggle with the growing rate of vaccine hesitancy identified among HCWs [20,21,22,23].

In our univariate analysis of these data, night shift work was associated with a reduced rate of decrease in the anti-SARS-CoV-2 antibody titer, and thus a higher level of protection among these HCWs. This finding was somewhat unexpected, given that a high risk of infection has been widely reported among NSWs compared to DWs [23]. However, this association was not significant upon subsequent multivariate analysis after controlling for study covariates; our findings suggest that the cardiometabolic frailty of the NSWs reported previously in several publications [24,25,26] may be associated with their overall susceptibility to SARS-CoV-2 infection [27,28,29].

Participants who were previously diagnosed with COVID-19 showed both stronger immunological responses and diminished loss of protective antibody levels over time compared to control (unaffected) subjects (2217.7 ± 575.6 U/mL versus 1066.0 ± 765.6 U/mL, respectively). This finding was identified as independent of all other relevant factors. This finding provides the research community with the opportunity to explore anamnestic responses to previous SARS-CoV-2 exposure among high-risk HCWs and may be particularly important given the high rate of both symptomatic and asymptomatic infections reported among these subjects [20,21]. Moreover, our results are also consistent with data reported previously in several publications that documented a significant relationship between the peak response and the persistence of protective antibody titers against many viral infections, including hepatitis B virus (HBV) [30,31,32].

Our findings also reveal that antibody production in response to vaccination was significantly influenced by participant age and sex. Compared to the participants who were both younger and female, the older male participants maintained significantly lower levels of antibody titers, albeit with similar rates of waning over time. Interestingly, recent results have revealed that while the duration of the antibody response after natural virus infection was somewhat variable, even in younger participants [33], there was a statistically significant relationship between age and the levels of the neutralizing antibody titer detected. Accordingly, results from a recent paper documented higher vaccine-induced anti-spike-RBD IgG in younger (ages 25–50 years) versus older (51–70 years) HCWs (*n* = 255) [34]. In another study (*n* = 33), serum antibody levels elicited by the mRNA vaccine were higher among younger participants, although this difference did not achieve statistical significance [35].

Our work has several limitations. First, our study mainly used a retrospective approach and had an exploratory nature, and no prospective analysis has been performed to date. Another limitation of this study was the relatively low number of participants who underwent multiple serological evaluations. This presents the possibility of selection bias as this activity might be undertaken more frequently by participants who developed comparatively low antibody titers immediately after vaccination. However, in our dataset, we found that the initial antibody titers measured in participants who underwent multiple evaluations were not significantly lower than the levels reported in those who underwent only one determination. Another limitation was that we did not address the presence and the magnitude of specific memory B and T cell responses. Robust adaptative responses were previously reported following the administration of the COVID-19 mRNA vaccine, although the relationship between these findings and the serological response remains unclear. Moreover, we did not determine the absence or presence of serum neutralizing capacity; thus, we are unable to determine whether serum concentrations of anti-S-RBD antibodies correlate directly with this capacity. We do note that other published studies have documented that serum neutralizing activity correlates directly with circulating titers of anti-spike or anti-RBD IgG [6]. Finally, we planned to extend the observational time beyond 9 months to better understand whether the expected decline in antibody levels may affect the long-term risk of COVID-19 infection.

By contrast, one strength of our study is that we evaluated the IgG titer in a large sample of HCWs at different times following vaccination. This approach permitted us to perform a quantitative evaluation of both the trend and the main determinants of the diminished antibody response over time.

This work contributes to the existing knowledge that focuses on the kinetics of the anti-SARS-CoV-2 antibody decline by providing a long-term evaluation of the anti-S-RBD IgG in vaccinated HCWs. Further studies will be needed to explore the neutralizing potential of all anti-SARS-CoV-2 antibodies and their role in promoting long-term protection from COVID-19 infection in real-life settings, as well as the need for booster doses among fully vaccinated subjects.

## 5. Conclusions

Our results reveal detectable levels of circulating protective anti-S-RBD antibodies in 99.5% of the participants in our study that persist for up to 250 days after the second dose of the BNT162b2 vaccine. While antibody levels waned over time, we found that the rate of decrease in repeated evaluations was directly related to the source of the response, the time elapsed, and a previous diagnosis of COVID-19. Interestingly, although antibody titers clearly waned over time following vaccination, no cases of severe COVID-19 were diagnosed among the HCWs enrolled in our study. Collectively, these data underscore the importance of achieving high levels of vaccine coverage among HCWs. The need for booster doses, specifically for those at high risk of occupational exposure, should also be carefully addressed.

## Figures and Tables

**Figure 1 vaccines-10-00141-f001:**
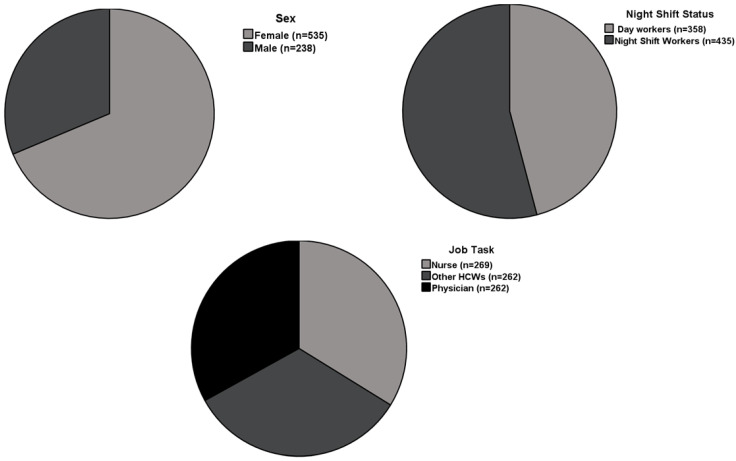
Study pie charts summarizing main characteristics of study participants.

**Figure 2 vaccines-10-00141-f002:**
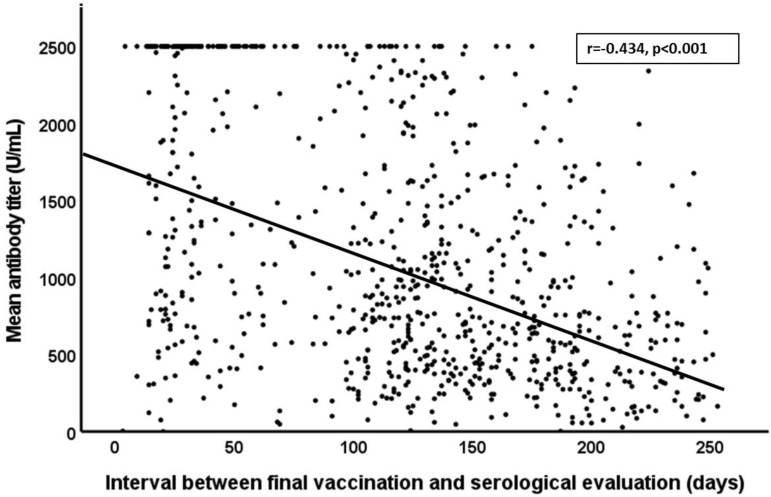
Correlation between mean antibody titer over time and days between final vaccination and serological evaluation.

**Figure 3 vaccines-10-00141-f003:**
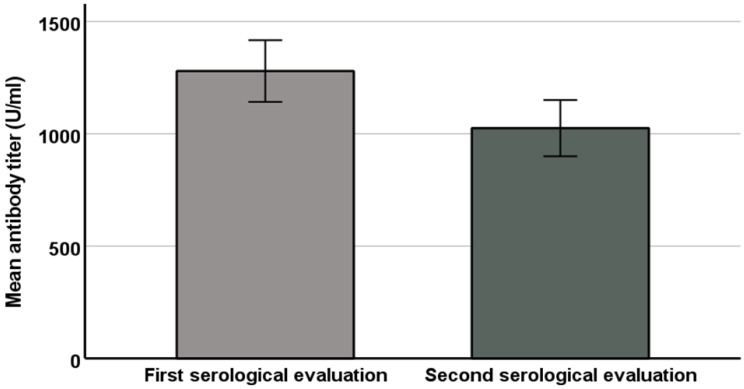
Mean antibody titer levels in first and second serological evaluation.

**Table 1 vaccines-10-00141-t001:** Characteristics of the study population including vaccination and serology status.

		N (%)	Mean ± Standard Deviation (SD)	Range
Age (years)			43.94 ± 11.30	21–77
Gender	Female	535 (67.50)		
	Male	258 (32.50)		
Serology for anti-S-RBD	Positive	791 (99.75)		
	Negative	2 (0.25)		
No. of serological evaluations	1	620 (78.19)		
	2	173 (21.81)		
Night shift status	Yes	435 (54.85)		
	No	358 (45.25)		
SARS-CoV-2 infection	Before vaccination	25 (3.15)		
	After vaccination	13 (1.65)		
Anti S-RDB titer (U/mL)	<90 days		1602.3	0–2500
	91–150 days		1035.9	0–2500
	>150 days		724.4	0–2500
Job task	Physician	262 (33.00)		
	Nurse	269 (33.90)		
	Other HCWs *	262 (33.00)		
Days elapsed between the final vaccine dose and the last serology evaluation			123 ± 65	13–253
Days between the two serology evaluations			92 ± 42	11–183

* Other HCWs include lab technicians, radiology technicians, perfusion technicians (including phlebotomists), neurophysiopathology technicians, administrative personnel, cardiocirculatory physiopathology technicians, psychologists, biologists, dieticians, dental hygienists, speech therapists, orthoptists, and pharmacists.

**Table 2 vaccines-10-00141-t002:** Factors associated with the observed decrease in anti-SARS-CoV-2 antibody titers in both univariate and multivariate analyses.

Variables	Univariate B Coefficient (95% CI)	*p*	Multivariate B Coefficient (95% CI)	*p*
Days between two serology tests	4.08 (2.46–5.70)	<0.01	3.21 (1.85–4.58)	<0.01
Night shift status	−199.74 (−330.00–−69.24)	<0.05	−92.67 (−208.00–22.65)	n.s.
Previous infection	−168.50 (−368.18–−31.06)	<0.05	−428.31 (−617.22–−240.63)	<0.01
First antibody titer	0.20 (0.13–0.27)	<0.01	0.248 (0.18–0.31)	<0.01
Age	−4.11 (−10.87–2.64)	n.s.	-	-
Male gender	−18.33 (−181.44–144.78)	n.s.	-	-

## Data Availability

Not applicable.

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
