# Peer review of "Reduced Titers of Circulating Anti-SARS-CoV-2 Antibodies and Risk of COVID-19 Infection in Healthcare Workers during the Nine Months after Immunization with the BNT162b2 mRNA Vaccine"

_vaccines, 2022, doi:10.3390/vaccines10020141_

Round 1
Reviewer 1 Report
Rizza and coauthors reported a retrospective study aimed to evaluate the determinants of serological protection in a group of vaccinated healthcare workers (HCWs) by evaluating circulating levels of anti-viral spike receptor-binding domain (S-RBD) antibodies during the nine-month period following vaccination. Their results revealed detectable levels of circulating protective anti-S-RBD antibodies in 99.5% of the participants in their study that persist for up to 250 days after the second dose of the BNT162b2 vaccine. The authors found that the rate of decrease of antibody levels in repeated evaluations was directly related to the source of the response, the time elapsed, and a previous diagnosis of COVID-19. Moreover, their data under-scored the importance of achieving high levels of vaccine coverage among HCWs. This work is meaningful and important for addressing the need for booster doses. Therefore, I recommend the paper be accepted by vaccines after addressing the following issues in a minor revision.
- The authors summarized the characteristics of the study population including vaccination and serology status in Table 1. However, it would be better if the authors could include some pie charts for the key data (such as gender, job task) so that the readers will have a better understanding.
- The authors stated that “Participants who were previously diagnosed with COVID-19 showed both stronger immunological responses and diminished loss of protective antibody levels over time compared to control (unaffected) subjects.” Could the authors include the values of protective antibody levels for the participants who were previously diagnosed with COVID-19 in this manuscript?
- The authors didn’t clarify the participants who were previously diagnosed with COVID-19 belonged to symptomatic or asymptomatic infections. As we know there might be some difference in immunological responses if a patient had a symptomatic infection. It would be better if the authors could further clarify it.
Author Response
Reviewer 1
Rizza and coauthors reported a retrospective study aimed to evaluate the determinants of serological protection in a group of vaccinated healthcare workers (HCWs) by evaluating circulating levels of anti-viral spike receptor-binding domain (S-RBD) antibodies during the nine-month period following vaccination. Their results revealed detectable levels of circulating protective anti-S-RBD antibodies in 99.5% of the participants in their study that persist for up to 250 days after the second dose of the BNT162b2 vaccine. The authors found that the rate of decrease of antibody levels in repeated evaluations was directly related to the source of the response, the time elapsed, and a previous diagnosis of COVID-19. Moreover, their data underscored the importance of achieving high levels of vaccine coverage among HCWs. This work is meaningful and important for addressing the need for booster doses. Therefore, I recommend the paper be accepted by vaccines after addressing the following issues in a minor revision.
Our response: we thank very much Rev1 for her/his positive and very appreciated comments.
The authors summarized the characteristics of the study population including vaccination and serology status in Table 1. However, it would be better if the authors could include some pie charts for the key data (such as gender, job task) so that the readers will have a better understanding.
Our response: as requested we created some pie charts to better display the characteristics of study population (Figure 1)
The authors stated that “Participants who were previously diagnosed with COVID-19 showed both stronger immunological responses and diminished loss of protective antibody levels over time compared to control (unaffected) subjects.” Could the authors include the values of protective antibody levels for the participants who were previously diagnosed with COVID-19 in this manuscript?
Our response: as wisely suggested by the Rev1, report regarding the protective value of antibody levels in subjects diagnosed with COVID-19 will provide valuable information. Accordingly, we added this information to discussion (page 9, lines 275-276)
The authors didn’t clarify the participants who were previously diagnosed with COVID-19 belonged to symptomatic or asymptomatic infections. As we know there might be some difference in immunological responses if a patient had a symptomatic infection. It would be better if the authors could further clarify it.
Our response: we thank again Rev1 for her/his usefull comment. Accordingly, we added the requested information among study results (page 3, lines 137-139)
Reviewer 2 Report
This manuscript is a retrospective summary and analysis of historical data. Because it is not well designed, it will affect the results and conclusions without considering whether there are new findings and results.
- The main result of the study is immunogenicity. Without the control group, the evaluation of protective effect cannot be seen, so the title is also inappropriate: “no impact on the protection of health workers”.
- Evaluation of the vaccine in real-world also needs control key factors such as sampling time, comparable immunogenicity evaluation methods, etc. which is not explained or controlled in the manuscript.
The observation time should be extended. At present, it is 9 months at most. The decline in antibody levels is expected, and many other studies have also reported. The distribution of observation time: the days elapsed between the final vaccine dose and the last serology evaluation and days between the two serology evaluations, all the scope is too large (table1). Take one or two samples per person at most, and the detection time is not uniform.
- The antibody detection reagent used in the manuscript can only reflect the immunogenicity of the vaccine, and the neutralizing antibody detection method should be considered. What is the correlation between the reagent used and the protective effect or neutralizing antibody? Please make a supplementary explanation.
- The results of the manuscript are not fully displayed. Only the table is used, the dynamic changes of antibodies cannot be seen, and the differences between different groups cannot be intuitively displayed. It should be displayed by graph.
Author Response
Reviewer 2
This manuscript is a retrospective summary and analysis of historical data. Because it is not well designed, it will affect the results and conclusions without considering whether there are new findings and results.
Our response: Rev 2 declares that the results and conclusion of our study are inconclusive because it is not well delineated due to its retrospective design. We agree with the reviewer and we thank her/him for this clever comment. Accordingly, we added this point as the first limitation of the study (page 9, lines 296-298).
The main result of the study is immunogenicity. Without the control group, the evaluation of protective effect cannot be seen, so the title is also inappropriate: “no impact on the protection of health workers”.
Our response: we agree with Rev2; accordingly, we changed the manuscript’s title
Evaluation of the vaccine in real-world also needs control key factors such as sampling time, comparable immunogenicity evaluation methods, etc. which is not explained or controlled in the manuscript.
Our response: we thank Rev2 for her/his clever comment. Actually, the best evaluation of vaccination efficacy is possible in intervention trials, whose study plan is created according to study endpoints. However, even if this is a retrospective study, we reported the sampling time in results as well in Table 1. Specifically, the days between the two serology evaluations were 92 ± 42 days (range 11 – 183) (page 4, lines 159-160)
The observation time should be extended. At present, it is 9 months at most. The decline in antibody levels is expected, and many other studies have also reported. The distribution of observation time: the days elapsed between the final vaccine dose and the last serology evaluation and days between the two serology evaluations, all the scope is too large (table1). Take one or two samples per person at most, and the detection time is not uniform.
Our response: we thank again Rev2 for her/his suggestion. Therefore, we have planned to extend the observation time beyond 9 months. Accordingly, we added this point to limitations (page 10, lines 312-314)
The antibody detection reagent used in the manuscript can only reflect the immunogenicity of the vaccine, and the neutralizing antibody detection method should be considered. What is the correlation between the reagent used and the protective effect or neutralizing antibody? Please make a supplementary explanation.
Our response: we are in perfect agreement with the Rev2 comment. Accordingly, we commented this limitation in page 10, lines 308-312
The results of the manuscript are not fully displayed. Only the table is used, the dynamic changes of antibodies cannot be seen, and the differences between different groups cannot be intuitively displayed. It should be displayed by graph.
Our response: in accordance with Rev2’s suggestion, we created a couple of ghraphs that now better display the dynamic changes of antibodies levels between groups. Ghraphs are now inserted in manuscript (Figure 2 and 3) with specific indications throughout the text.
Round 2
Reviewer 2 Report
The author has revised and supplemented the discussion according to the comments of the reviewer.